# Genomic and Transcriptomic Predictors of Response to Immune Checkpoint Inhibitors in Melanoma Patients: A Machine Learning Approach

**DOI:** 10.3390/cancers14225605

**Published:** 2022-11-15

**Authors:** Yaman B. Ahmed, Ayah N. Al-Bzour, Obada E. Ababneh, Hassan M. Abushukair, Anwaar Saeed

**Affiliations:** 1Faculty of Medicine, Jordan University of Science and Technology, Irbid 22110, Jordan; 2Department of Medicine, Division of Medical Oncology, Kansas University Cancer Center, Kansas City, KS 66205, USA; 3Department of Medicine, Division of Hematology and Oncology, UPMC Hillman Cancer Center, University of Pittsburgh, Pittsburgh, PA 15213, USA

**Keywords:** melanoma, immune checkpoint inhibitors, machine learning, tumor mutational burden

## Abstract

**Simple Summary:**

Our work provides novel transcriptomic biomarkers that can accurately predict immune checkpoint inhibitors (ICIs) response in melanoma patients. Using a bioinformatics analysis and supervised machine learning approach, we developed four random-forest classifiers based on clinical, genomic, transcriptomic and survival data. The results of these models can enable further insight into the potential role of these genes in immunotherapy. In addition, our findings were based on a supervised approach, in which melanoma patients treated with ICI were used to retrieve response-associated biomarkers, unlike several studies that used an unsupervised approach based on drug targets to predict ICI response in non-ICI-treated melanoma patients. Apart from ICI response, we also investigated the effect of these biomarkers on overall survival and patients’ prognosis, which also revealed a high association with survival, marking these biomarkers as powerful in both ICI response and patients’ prognosis. Thus, our work demonstrates a cornerstone in precision oncology and further evaluates these biomarkers in clinical practice using personalized medicine for a better prognosis and response outcomes.

**Abstract:**

Immune checkpoint inhibitors (ICIs) became one of the most revolutionary cancer treatments, especially in melanoma. While they have been proven to prolong survival with lesser side effects compared to chemotherapy, the accurate prediction of response remains to be an unmet gap. Thus, we aim to identify accurate clinical and transcriptomic biomarkers for ICI response in melanoma. We also provide mechanistic insight into how high-performing markers impose their effect on the tumor microenvironment (TME). Clinical and transcriptomic data were retrieved from melanoma studies administering ICIs from cBioportal and GEO databases. Four machine learning models were developed using random-forest classification (RFC) entailing clinical and genomic features (RFC7), differentially expressed genes (DEGs, RFC-Seq), survival-related DEGs (RFC-Surv) and a combination model. The xCELL algorithm was used to investigate the TME. A total of 212 ICI-treated melanoma patients were identified. All models achieved a high area under the curve (AUC) and bootstrap estimate (RFC7: 0.71, 0.74; RFC-Seq: 0.87, 0.75; RFC-Surv: 0.76, 0.76, respectively). Tumor mutation burden, GSTA3, and VNN2 were the highest contributing features. Tumor infiltration analyses revealed a direct correlation between upregulated genes and CD8+, CD4+ T cells, and B cells and inversely correlated with myeloid-derived suppressor cells. Our findings confirmed the accuracy of several genomic, clinical, and transcriptomic-based RFC models, that could further support the use of TMB in predicting response to ICIs. Novel genes (GSTA3 and VNN2) were identified through RFC-seq and RFC-surv models that could serve as genomic biomarkers after robust validation.

## 1. Introduction

Immune checkpoint inhibitors (ICIs) have revolutionized the treatment of advanced melanoma. These agents work by decreasing the inhibition of the immune cell, thus, facilitating an immune response against the tumor. Such agents include pembrolizumab and nivolumab (anti-PD-1), atezolizumab (anti-PDL-1) and ipilimumab (anti-CTLA-4). These ICIs are approved by the US Food and Drug Administration (FDA) for the treatment of advanced melanoma [1]. Although the ICIs published studies revealed favorable initial response rates and lasting responses to immunotherapy, about 50% of patients do not experience clinical response [2]. The need for validated biomarkers to detect the progress of a clinically prominent antitumor immune response in treated patients is also there [3].

Different predictors of the response to immunotherapy have been proposed. PDL-1 expression on tumor cells pre-treatment is most intricately linked with response to PD-1 inhibitors [4]. The tumor microenvironment (TME) involves the infiltration of both immune cells (tumor immune microenvironment [TIME]) and other mesenchymal cells [5]. The characteristics of TME are linked to response and resistance to ICIs in melanoma [6]. It is noted that the prediction of ICI response based on multiple genomic, immunogenic, and clinical variables is a prominent goal of current cancer-related research.

Since the prediction of response to ICI therapy is an important factor in devising a treatment plan for cancer patients, the introduction of machine-learning-based models has been suggested to help in patient selection and improve patients’ prognosis. Machine learning algorithms can predict responses to ICIs by integrating genomic, molecular, demographic, and clinical data [7]. These models work by encompassing several variables in a complementary manner to act as training parameters to predict patients’ responses to treatment [8]. Our goal is to integrate a variety of demographic, clinical, genomic, and transcriptomic variables to predict the ICI response and understand the primary resistance mechanisms in melanoma patients using a machine learning classifier.

## 2. Materials and Methods

### 2.1. Data Acquisition

We retrieved the clinical data of immunogenomic studies on melanoma patients who underwent ICI therapy through the cBioportal database (https://www.cbioportal.org) up to May 2022, [9] an integrated web tool for clinical and cancer genomic data. A total of 8 datasets accounted for immunogenomic studies in the cBioportal database, 3 datasets were on ICI-treated melanoma patients [10,11,12]. The retrieved clinical data included the following variables: melanoma histological subtype, age, sex, ICI agent, tumor mutational burden (TMB), the fraction of the genome altered (FGA), mutation count, and treatment response. Tumor objective response was used to classify patients as responders if they showed complete response (CR) or partial response (PR), and as non-responders, if they showed stable disease (SD) or progressive disease (PD) according to the Response Evaluation Criteria In Solid Tumors (RECIST 1.1) [13]. Additionally, the Gene Expression Omnibus (GEO) database (https://www.ncbi.nlm.nih.gov/geo/), was inspected for gene expression profiles of melanoma patients treated with ICIs up to May 2022. Figure 1 illustrates data workflow.

### 2.2. Differential Expression Analysis (DEA)

The GSE91061 dataset of high-throughput sequencing was used to screen out the differentially expressed genes (DEGs) associated with ICI responses between responders and non-responders in 109 melanoma patients. DEA was performed using DESeq2 from Bioconductor R package (v3.13), which uses Bayes shrinkage to estimate dispersion and fold-change (FC) then fit a generalized linear model (GLM) for each gene based on a negative binomial distribution, and tests for significance using the Wald-test [14]. We set a *p*-value threshold < 0.05 corrected for multiple comparisons using the Benjamini and Hochberg method. A cutoff point of |log2FC| > 0.5 was set to determine the up- and down-regulated DEGs.

### 2.3. Immune Infiltration Analysis

The xCELL algorithm, a web-based tool for enrichment analysis based on gene expression profiles, was used to estimate the immune scores of 64 immune components and stromal cells and provides an understanding of cellular heterogeneity across tumor tissue [15]. Hierarchical clustering was used to assess the correlation between the 21 most relevant immune components and stromal cells and the DEGs with |log2FC| > 2.0 using the pheatmap package from R gplots (v1.0.12; https://CRAN.R-project.org/package=pheatmap) and to assess the effect of gene expression and the tumor immune microenvironment.

### 2.4. Gene Ontology (GO) Enrichment Analysis

To understand the functional landscape of the 723 upregulated and 303 downregulated DEGs, we performed a GO enrichment analysis for biological processes (BP), molecular function (MF), and cellular components (CC). Using the clusterprofiler package from Bioconductor R, an adjusted *p*-value < 0.05 threshold was considered statistically significant for GO enrichment terms.

### 2.5. Support Vector Machine—Recursive Feature Elimination (SVM-RFE)

The 1066 DEGs were fitted into an SVM model to predict ICI response, then RFE was used to select the top 5 ranking genes in predicting ICI response. The RFE is a backward-iterative approach that eliminates features that do not show a correlation with the target variable, which enhances the accuracy of the model’s prediction. RFE calculates the ranked weight of all features and sorts them based on their classification performance [16]. The top-four-ranking upregulated genes and the top-four-ranking downregulated genes were then fitted into a random-forest classifier to evaluate their prediction performance.

### 2.6. Survival-Associated DEGs

To evaluate the effect of ICI-predictor DEGs on melanoma prognosis, we recruited the top 100 ranking DEGs in predicting ICI response from the SVM-RFE model. To select genes with both prognostic and predictive values, the top 100 DEGs were cross-validated with the survival-related genes in melanoma patients of the TCGA cohort, which were obtained from the GEPIA2 database. The overlapping genes from the top 100 ICI-predicting DEGs and survival-related genes were fitted into a Cox proportional hazard model (CPH) to evaluate its effect on OS from 3 datasets which were obtained from GEO and cBioportal databases, including GSE78220 (n = 27), Metastatic Melanoma, DFCI 2015 (n = 40) 11, Melanoma, MSKCC 2014 (n = 21) 12. A best-cutoff point for age and the overlapping genes was set by computing a cox model over the values between the first and third quartiles (Q3-Q1) for each variable, choosing the most significant cutoff value, and separating the variables into two groups. Multiple hypotheses were generated during the calculation of the best-cutoff value; thus, we corrected the *p*-values for multiple comparisons using the “BH” method. The survival and rms package from R were used to carry out the analyses.

### 2.7. Random Forest Classifier

Three machine-learning random-forest classifications (RFC) ensembles were built to predict ICI responses in the GSE91061 and GSE78220 datasets of 133 ICI-treated melanoma patients. The first RFC model (hereafter called RFC-Seq) was trained on the 8 top ranking up- and downregulated DEGs from the SVM-RFE model with a training-testing ratio of 8:2, in which 108 patients contributed to the training set, and 25 patients contributed to the validation set. The second RFC model (hereafter called RFC-Surv) was trained on the 8 survival-related genes from the CPH model, whereas the third RFC model (hereafter called RFC16) was trained on 16 features from the RFC-Seq and RFC-Surv models. All models’ performances were evaluated on the testing set using the mean bootstrap estimate with a 95% confidence interval (95% CI), area under the receiver operating characteristics curve (AUC/ROC), and 10-fold cross-validation in which the whole dataset of 133 patients was split into 10 iterations with 9 folds of the data for the training and the remaining fold for testing in each iteration, then calculating the mean score from each iteration. The scikit-learn package from python was used for model construction and evaluation [17,18]. To assess the contribution of each gene in predicting ICI response, feature contribution was calculated using the permutation importance which represents the decrease in the fitted model score when the values of each feature are randomly shuffled, thus, a higher decrease in the model’s score indicates a higher dependence on this feature in predicting the target.

### 2.8. Data Availability

The data used in this study are publicly available from the gene expression omnibus (GEO) and cbioportal databases can be accessed through GSE91061 and GSE78220. The study workflow is shown in Figure 1.

## 3. Results

### 3.1. Clinical Data and ICI Response Model

A total of 212 ICI-treated melanoma patients were recruited from the cBioPortal database. Age at diagnosis was partially significant between R and NR (*p*-value = 0.04), while TMB differed significantly (mean log2 (TMB+1): 3.14 versus 4.03, *p*-value < 0.0001). FGA and sex were not significantly different between R and NR. A random-forest classifier (RFC) was trained to predict ICI response based on 7 clinical features (hereafter called RFC7), including log2 (TMB+1), FGA, mutation count, sex, age at diagnosis, ICI agent, and histological subtype. The RFC7 model performed on the testing set with a mean bootstrap estimate of 0.74 with 95% CI: [0.66–0.81], 10-fold cross-validation of 0.66, and AUC of 0.71 as shown in Figure 2F. Demographic and clinical characteristics are shown in Table 1. TMB was the highest contributing feature in predicting ICI response, followed by mutation count, FGA, and age as shown in Figure 2A. To evaluate the predictive performance of TMB alone in ICI response, an RFC model was trained on TMB alone and performed on the testing set with a mean bootstrap estimate of 0.67 and 95% CI: [0.50–0.84], 10-fold cross-validation of 0.61, and AUC of 0.68, thus it did not show better performance compared to the RFC7 model as shown in Figure 2E.

### 3.2. Differentially Expressed Genes (DEGs) and RFC-Seq Model

A total of 22,187 genes were screened out for differential expression between R and NR. The DEA resulted in 1066 DEGs, 763 of which were upregulated (log_2_FC > 0.5) while 303 genes were downregulated (log_2_FC < −0.5). Figure 3A shows a volcano plot for the DEGs. The resulting DEGs were trained on the SVM-RFE model to select the top 4 upregulated and top 4 downregulated ranking DEGs. IGHD, SIGLEC8, CLDN14, and CLU were the top-four-ranking upregulated DEGs, while CORO2B, EHF, GSTA3, and ANXA3 were the top-four-ranking downregulated DEGs in predicting ICI response, and they were then fitted into the RFC-Seq. The RFC-Seq model was fitted into the resulting DEGs from GSE91061 and GSE78220 datasets with a training-validation ratio of 8:2. The RFC-Seq model performed on the testing set with a mean bootstrap estimate of 0.75 and 95% CI: [0.54–0.93], 10-fold cross-validation of 0.67 and AUC of 0.87. Table 2 shows the evaluation metrics for the RFC-Seq model. Figure 2F shows the ROC curve for the RFC-Seq model. GSTA3 was the highest contributing feature as shown in Figure 2B, followed by CLDN14. An RFC model was solely trained on GSTA3 to evaluate its performance in predicting ICI response alone and performed on the testing set with a higher mean bootstrap estimate of 0.83 and 95% CI: [0.62–1.00], 10-fold cross-validation of 0.65, and AUC of 0.80, as shown in Figure 2E. It showed better certainty in IC predicting; however, it did not show better overall sensitivity or specificity.

### 3.3. Tumor Immune Microenvironment and DEGs

The correlation matrix between 21 immune components and 144 DEGs in ICI responders revealed a significant inverse correlation between 31 upregulated genes including OLIG1, UBAP1L, CLDN14, WNK4, and myeloid-deprived suppressor cells (MDSCs) (r = −0.5, 0.6, and −0.7, respectively, *p*-value < 0.05). 31 upregulated DEGs showed significant direct correlation with B-cells and CD4+ cells, including: CCR6, CCR7, CR2, CD97A, CD40LG, CD5, FCRL5, TLR10, SPIB, and CCL21 (r > 0.9, *p*-value < 0.0001). CD8+ cells were significantly correlated with 28 upregulated DEGs, including DTX1, CR2, FCER2, and LILRA4 (r > 0.7, *p*-value < 0.001). Six downregulated DEGs showed a significant direct correlation with neutrophils (r > 0.6, *p*-value < 0.05), including EHF, SFN, ANXA3, TACSTD2, S100A14, and S100A8. While 2 downregulated DEGs showed a direct correlation with melanocytes including WNK4 and CCD140 (r > 0.6, *p*-value < 0.05). Figure 3B presents a dendrogram heatmap for the correlation between the tumor immune microenvironment and DEGs. Figure 4 shows a heatmap of the genomic and immune landscape between responders and non-responders.

### 3.4. Gene Ontology (GO) Enrichment Analysis

The enrichment analysis of GO in the 723 upregulated DEGs showed significant enrichment in BP including: “T-cell activation”, “mononuclear cell differentiation”, “lymphocyte differentiation”, “regulation of T-cell activation”, and “leukocyte cell–cell adhesion”. MF was significantly enriched in “MHC protein complex binding”, “immune receptor activity”, “MHC- class II protein complex binding”, and “cytokine receptor activity”. GO terms for CC were significantly enriched in “vacuolar membrane”, “lysosomal membrane”, “immunological synapse”, and “MHC protein complex”. While GO terms in BP for the 303 downregulated DEGs were significantly enriched in: “keratinization”, “skin development”, “keratinocyte differentiation”, and “epidermal cell differentiation”. While “structural constituent of skin epidermis” and “cornified envelope” were the only GO terms enriched in MF and CC, respectively. Figure 5A,B show GO for BP, MF, and CC for the upregulated and downregulated DEGs, respectively.

### 3.5. Survival-Associated DEGs and RFC-Surv Model

Of the 1066 DEGs, the top 100 DEGs in predicting ICI response from the SVM-RFE model are shown in Figure 1. A total of 500 survival-related genes were identified from the GEPIA2 database in melanoma patients from the SKCM-TCGA cohort. The following eight genes CCL5, IL4I1, VNN2, PARP15, LCK, ZNF831, CD86, and FCRRL6 were overlapping between the SKCM-TCGA and top 100 ICI-predicting DEGs as illustrated in Venn Diagram Figure 2D. The eight overlapping genes were included in the univariate cox regression model in addition to age and sex. All the genes were significantly associated with survival (*p* < 0.05) as shown in Table 3. Those genes were then studied for confounding effects using the multivariate cox regression. None of the genes showed an independent association with OS when accounting for covariates (*p*-value > 0.05). Figure 6A shows a Kaplan–Meier plot for the OS between high and low groups of OS-related genes and a forest plot for the CPH model in Figure 6B. To further test their prediction in ICI response, those genes were fitted into the RFC-Surv model and performed on the testing set with 0.76 mean bootstrap estimate, 95% CI: [0.54–0.91], 10-fold CV of 0.68 and AUC of 0.76. VNN2 showed the highest feature importance, followed by FCRL6, IL4I1, PARP15, and ZNF831, as shown in Figure 2C, and the expression of all genes was higher in responders compared to non-responders (Mann–Whitney test *p*-value < 0.05) as shown in Figure 7. To evaluate VNN2 performance in predicting ICI response alone, it was fitted into an RFC model and performed on the testing set with a higher mean bootstrap estimate of 0.84 and 95% CI: [0.63–1.00], 10-fold cross-validation of 0.70, and AUC of 0.82 as shown in Figure 2E, thus showing a better overall accurate performance in predicting ICI response.

The features of the RFC-Seq model (n = 8) and the RFC-Surv model (n = 8) were fitted into a combined model (RFC16) and performed on the testing set with 0.77 mean bootstrap estimate, 95% CI: [0.58–0.92], 10-fold CV of 0.67 and AUC of 0.88 as shown in Table 2 and Figure 2F. VNN2 and GSTA3 were the highest contributing features in the RFC16 model as shown in Figure 5C.

## 4. Discussion

Accurate prediction of immunotherapy response in cancer patients remains an elusive question. Several studies have investigated the potential effect of multiple demographic, clinical, and genomic variables in the prediction of immunotherapy response. Herein, through utilizing melanoma open data-driven cohorts, we built a machine learning classifier to integrate demographic, genomic as well as transcriptomic data to predict response to ICIs. Our RFC7 model further confirmed the substantial role of TMB in predicting responders to ICIs in melanoma as it was the highest contributing feature.

The RFC TMB alone model did not perform better in comparison to the RFC7 model which is in agreement with a previous machine learning report in solid cancers where a multifactorial model including both genomic and clinical features outperformed the TMB-based model [19]. In their study, Chowell and colleagues developed a machine learning model using 1479 patients across 16 cancer types (37% had non-small cell lung cancer and 13% had melanoma) [19]. Their 16-variable model which incorporated clinical (age, sex, ICI agent, cancer stage, BMI, neutrophil-to-lymphocyte ratio, cancer type, chemotherapy status, blood parameters), molecular and genomic (TMB, fraction of copy number alterations, HLA-1 evolutionary divergence, loss of heterozygosity in HLA-1, MSI) has outperformed the established TMB based model (AUC: 0.85 versus 0.62). Other established ICI response biomarkers include CD8+ T cell infiltration levels and PD-L1 expression levels both on tumor and immune cells [20,21]. One study further highlighted the prognostic value of the presence of CD137 on circulating CD8+ T cells in resected stage 3 melanoma patients after adjuvant ipilimumab + nivolumab combination therapy [22]. Our results should be confirmed in clinical trial settings to assess their performance in comparison to other reported predictors.

TMB remains to be the only established biomarker for ICI response as it was approved by the FDA as a predictive biomarker for response in solid tumors patients treated with anti-PD-1 pembrolizumab following phase II of KEYNOTE-158. [23,24]. A suggested mechanism is that high TMB generates multiple neo-peptides of immunogenic cancer cells, thus being recognized by T cells when presented by MHC molecules and enhancing the response to ICIs [25]. One of the challenges regarding the use of TMB as an optimal biomarker for ICI prediction is the absence of a fixed tumor-type-specific TMB cutoff [26]. The universal 10 mut/Mb cutoff to define high TMB showed high variation in its accuracy as a precision immune-oncology tool across cancer types with higher power of prediction seen in melanoma and NSCLC patients and lower power in other solid and renal cancers. Thus, to optimize the use of TMB in ICI blockade prediction, tailored cancer-specific TMB cutoffs are warranted [27,28]. TMB variability across tumors reflects the level of chronic exposure to carcinogens such as tobacco smoking in lung cancer and ultraviolet light in melanoma which are translated to higher TMB, thus contributing to better responses to ICIs blockade in lung cancer and melanoma [29,30,31]. The pathologic and molecular heterogeneity between various malignancies limits the function of TMB alone as a predictor of ICI response and raises the need of integrating other genomic and demographic features, which may influence ICI response prediction. FGA which represents the percentage of copy number alteration in a specific chromosomal region, and mutation count which represents the number of mutational events in each patient, are thought to be cancer-specific factors that influence disease progression, however, are not well-validated as predictive biomarkers of ICIs [32]. Kong et al. demonstrated that the combination of TMB with a network-based analysis that identifies ICI-target-proximal genes and pathways, improved the prediction of ICI response as well as overall survival in melanoma, further confirming the applicability of a combinatorial approach to multi-omics datasets to achieve superior predictive accuracy with ICIs [33].

Through transcriptomic analyses, we retrieved the significantly up- and downregulated genes which were then included as features in the RFC-seq model. The most contributing gene to the overall high-performance accuracy of the model was the glutathione S-transferase A3 (GSTA3) enzyme. Interestingly, the GSTA3-based RFC model had a numerically superior accuracy yet lower sensitivity and specificity in comparison to the RFC-seq model. GSTA3, a member of the glutathione S-transferase family, is involved in cellular defense against toxic carcinogenic compounds [34]. Previous studies have alluded to GSTA3’s involvement in tumorigenesis in colon cancer and gastric cancer as higher expression translated into poorer survival outcomes [35,36]. A recent report on cutaneous squamous cell carcinoma highlighted a tumor-suppressive effect on progression by inhibiting the TGF-beta/Smad and HIF-1alpha signaling pathways [37].

To further understand the immune landscape of responding patients, upregulated genes were found to be associated with higher infiltration levels of CD8+, CD4+ T cells, and B cells which are known to promote a durable response to ICIs in melanoma [38]. Interestingly, a recent study pointed out that the co-occurrence of CD20+ B cells and CD8+ T cells was independently associated with improved survival in melanoma patients. Additionally, the presence of B cells was associated with the formation of tertiary lymphoid structures which enable CD20+ B cells for antigen presentation and overall assistance in inducing durable immune responses to ICIs [39]. Among the 144 DEGs, 31 upregulated genes were found to be inversely associated with MDSCs which are one of the main immunosuppressive cells in the tumor microenvironment that promote tumor progression. Therefore, agents targeting MDSCs are one of the currently investigated strategies to overcome ICI resistance [40]. One of those agents, ATRA, currently being studied in phase 2 trials, has demonstrated the ability to induce MDSCs differentiation into macrophages and dendritic cells [41,42]. In addition, GO analyses revealed enrichment for immune activation terms such as “T-cell activation”, “MHC protein complex binding” and “immunological synapse” in the 723 upregulated genes in responders. This further confirms the unique immune microenvironment of melanoma responders. As for downregulated genes, enrichment with skin-related terms was observed “keratinization”, “skin development”, “keratinocyte differentiation”, and “epidermal cell differentiation”. This is supported by the spindle cell morphology in melanomas which represents a dedifferentiated mesenchymal phenotype, and epithelial to mesenchymal transition (EMT) which was shown to be directly correlated with PD-L1 expression [43,44].

The developed RFC-surv model using survival-related genes performed comparably with the RFC-seq model (mean bootstrap measure 0.76 versus 0.75) and VNN2 was identified as the main contributing feature. Further analyses on VNN2 through a single gene RFC model, demonstrated higher performance accuracy and superior specificity and sensitivity in comparison to the 8-gene RFC-surv model. VNN2, a member of the vanin family, is a membrane-associated protein mainly expressed on myeloid and lymphoid cells implicated in the inflammatory response [45]. VNN2 was overexpressed in ICI-responding patients in our sample and notably, its lower expression was associated with poor survival outcomes. In other solid tumors, Li et al. identified VNN2 as one of the six-gene signatures that predict overall survival in hepatocellular carcinoma in which higher VNN2 expression correlated with a poor prognosis [46]. While in hematological malignancies, Bornhauser et al. identified VNN2 as a marker that increases resistance to chemotherapy in acute lymphoblastic leukemia [47]. A urologic cancer-based study provided some mechanistic insight into VNN2’s impact, in which VNN2 was found to promote non-adhesive proliferation and IL-1beta production in prostate cancer cell lines and generally was correlated with poor survival outcomes [48]. This effect could be attributed to VNN2’s effect of inducing chronic inflammation through NF-KB activation which is known to suppress the immune response [48]. Another aspect of VNN2’s role in the tumor immune microenvironment is its functionality with MDSCs. One study has identified VNN2 as a unique surface enzyme highly enriched on monocytic MDSCs in healthy subjects [49]. While MDSCs are known to inhibit CD8+ T cell proliferation, this study described an inverse association between CD14+ monocytes expressing VNN2 and glioma patient’s tumor grade potentially highlighting a role for VNN2 in less aggressive tumors [49]. In another study on metastatic renal cell carcinoma, VNN2 expression on monocytes and neutrophils was found to be correlated with poor prognostic outcomes [50]. Considering the limited work pertaining to VNN2′s significance in melanoma, mechanistic studies describing its effect on the tumor immune microenvironment are needed to establish practical recommendations.

We also developed an RFC model that combines features from the RFC-seq and the RFC-surv model, to optimize our machine learning approach for more than one aspect of the genomic landscape of the microenvironment. Using the combined features model, VNN2 and GSTA3 remained to be the major contributing genes which further demonstrates those genes’ substantial involvement in response prediction and warrants future experimentation for biomarker validation.

In this study, we conducted a comprehensive bioinformatics analysis, integrating multiple genomic and demographic factors contributing to ICI prediction using robust machine learning models. Our results and in concordance with previous reports present a strong association between TMB and ICI responses. In addition, we provided a unique insight into the transcriptomic landscape of melanoma ICI responders. However, our models were limited by the relatively small sample size of the melanoma cohorts. Additionally, due to the lack of large melanoma cohorts with adequate genomic and transcriptomic data, we could not externally validate our models. Lastly, our cohorts did not include patients on anti-PD-1 plus anti-CTLA-4 combination therapy which limits the generalizability of our model on such treatment which has been incorporated in advanced melanoma first-line treatment. Therefore, validation of external cohorts with adequate sample sizes and inclusivity of ICI combination regimens are needed in the future.

## 5. Conclusions

We introduced and evaluated machine learning models incorporating several genomic and transcriptomic data that predicted responses to immunotherapy in advanced melanoma patients. Our results further confirmed the accuracy of TMB in predicting response to ICIs and demonstrated the applicability of integrating other clinical and genomic features to improve the performance of TMB in predicting response. Additionally, through transcriptomic data, we were able to identify novel genes (GSTA3 and VNN2) that could lead to better stratification of potential responders to ICIs in melanoma.

## Figures and Tables

**Figure 1 cancers-14-05605-f001:**
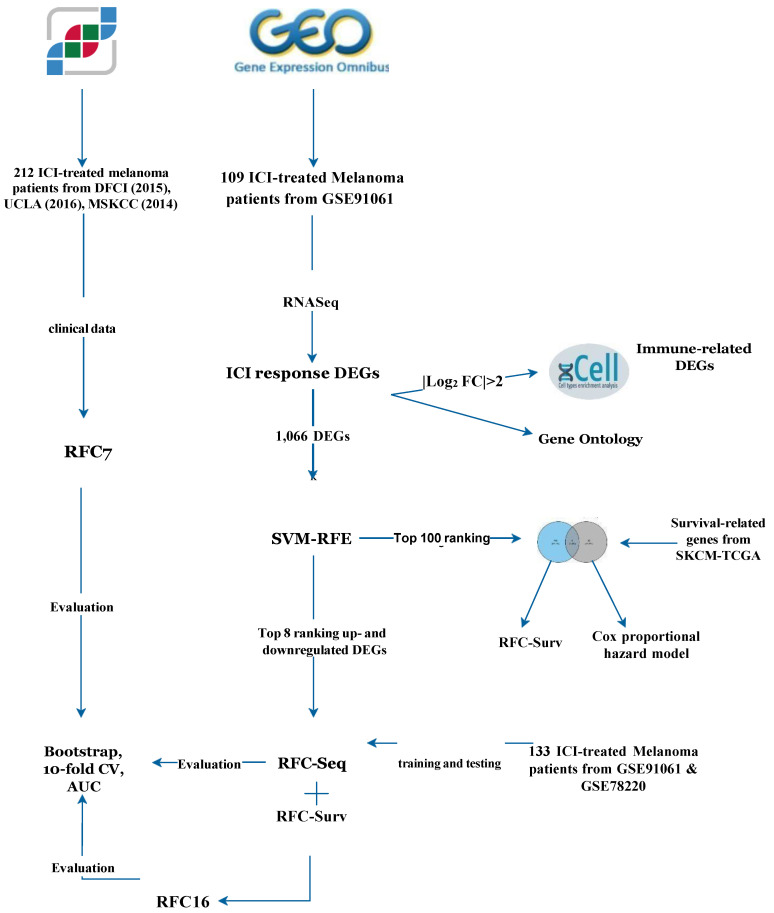
Workflow of the study. Four RFC models were built, a clinical and genomic model (RFC7) based on 3 cBioPortal cohorts, transcriptomic model (RFC-Seq) based on RNA-Seq in GSE91061, RFC-Surv based on survival-related genes and RFC16 based on genes in RFC-Seq and RFC-Surv models. DEGs of GSE91061 based on ICI response were further used to understand TME, GO, and to build the RFC-Seq model. The intersected genes between the top 100 immune-related genes identified via SVM-RFE and survival-related genes from SKCM-TCGA were used to identify prognostic role of these genes using Cox proportional hazard model and to build RFC-Surv model to predict response. Based on genes used in RFC-Seq and RFC-Surv, RFC16 was built to predict response to ICI. ICI: immune checkpoint inhibitors, DEGs: differential expressed genes, SVM-RFE: Support Vector Machine—Recursive Feature Elimination, SKCM-TCGA: The Cancer Genome Atlas- skin cutaneous melanoma, AUC: area under the curve.

**Figure 2 cancers-14-05605-f002:**
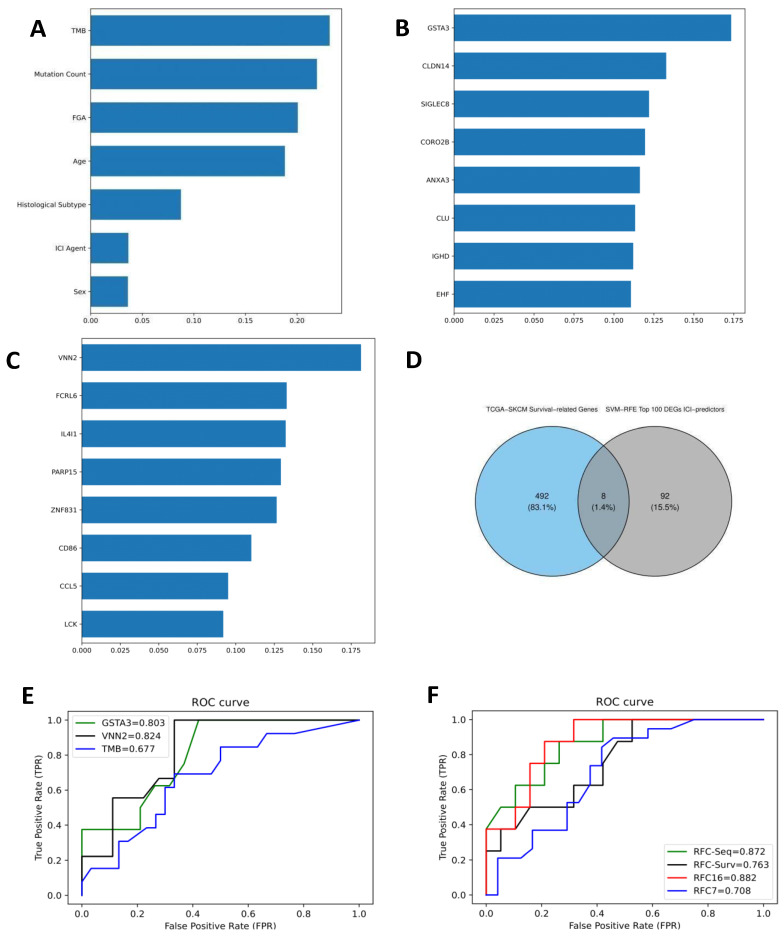
ICI prediction models: (**A**) Feature contribution of RFC7 model showing TMB as the strongest predictor of response. (**B**) Feature contribution of RFC-Seq model showing GSTA3 as the strongest predictor of response. (**C**) Feature contribution of RFC-Surv model showing VNN2 as the strongest predictor of response. (**D**) Venn diagram showing the intersection between TCGA−SKCM Survival−related Genes and SVM−RFE Top 100 DEGs ICI−predictors. (**E**) The ROC curve of the top associated features from RFC7, RFC-Seq and RFC-Surv models. (**F**) The ROC curve of RFC7, RFC-Seq and RFC-Surv models.

**Figure 3 cancers-14-05605-f003:**
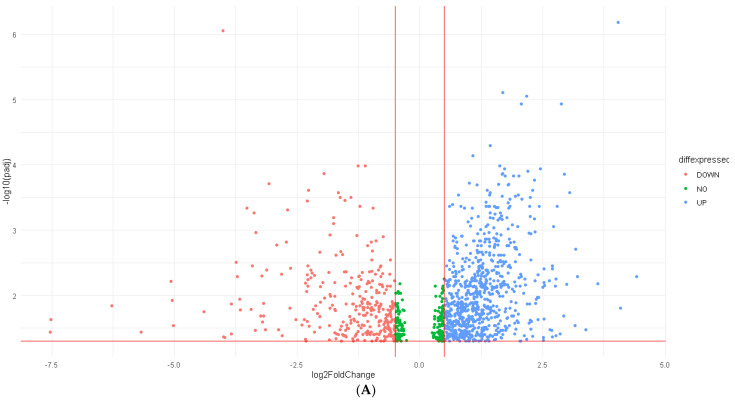
DEGs and the landscape of TME: (**A**) Volcano plot showing the DEGs based on response to ICI in GSE91061 as blue dots denote upregulated genes (log_2_FC > 0.5) while red dots denote downregulated genes (log_2_FC < −0.5). (**B**) Dendrogram of the DEGs and their association with the TME.

**Figure 4 cancers-14-05605-f004:**
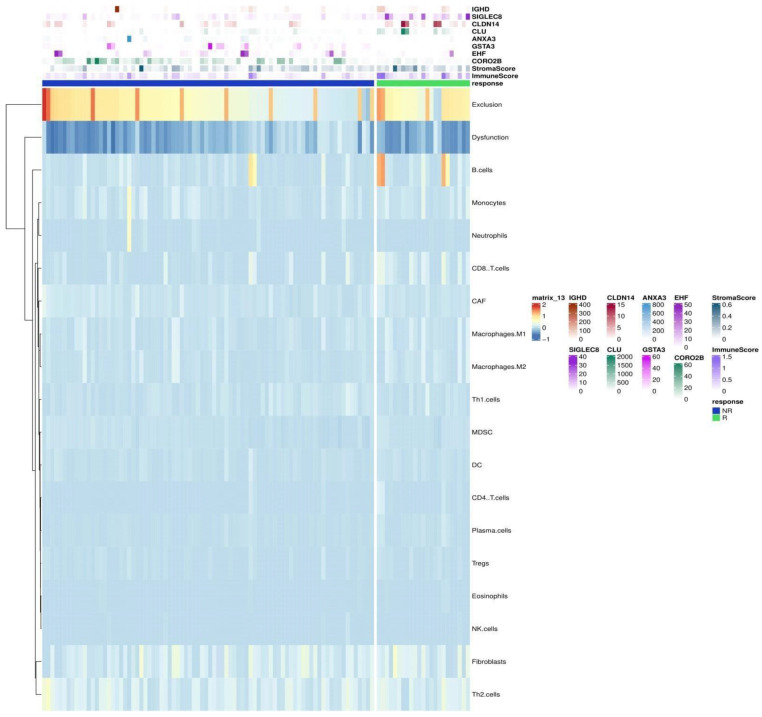
Heatmap of top DEGs and TME based on response to ICI. Top rows show the top 4 upregulated (IGHD, SIGLEC8, CLDN14, and CLU) and the top 4 downregulated (ANXA3, GSTA3, EHF, and CORO2B) genes based on SVM-RFE in the responders and non-responder patients’ groups. Rest of the rows show the landscape of TME.

**Figure 5 cancers-14-05605-f005:**
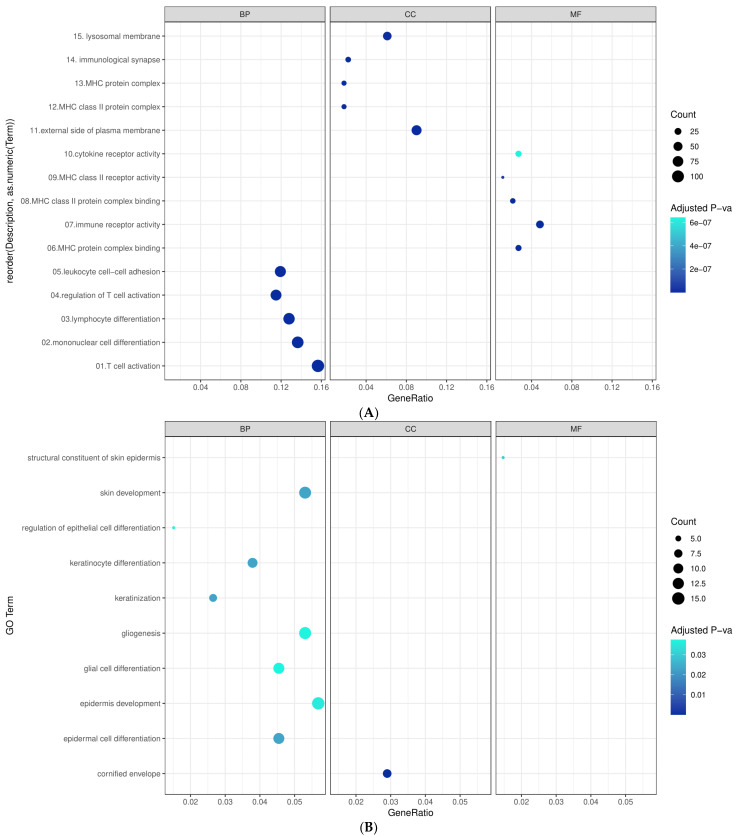
Functional enrichment analysis of the 1066 DEGs: (**A**) GO for the upregulated DEGs. (**B**) GO for the downregulated DEGs.

**Figure 6 cancers-14-05605-f006:**
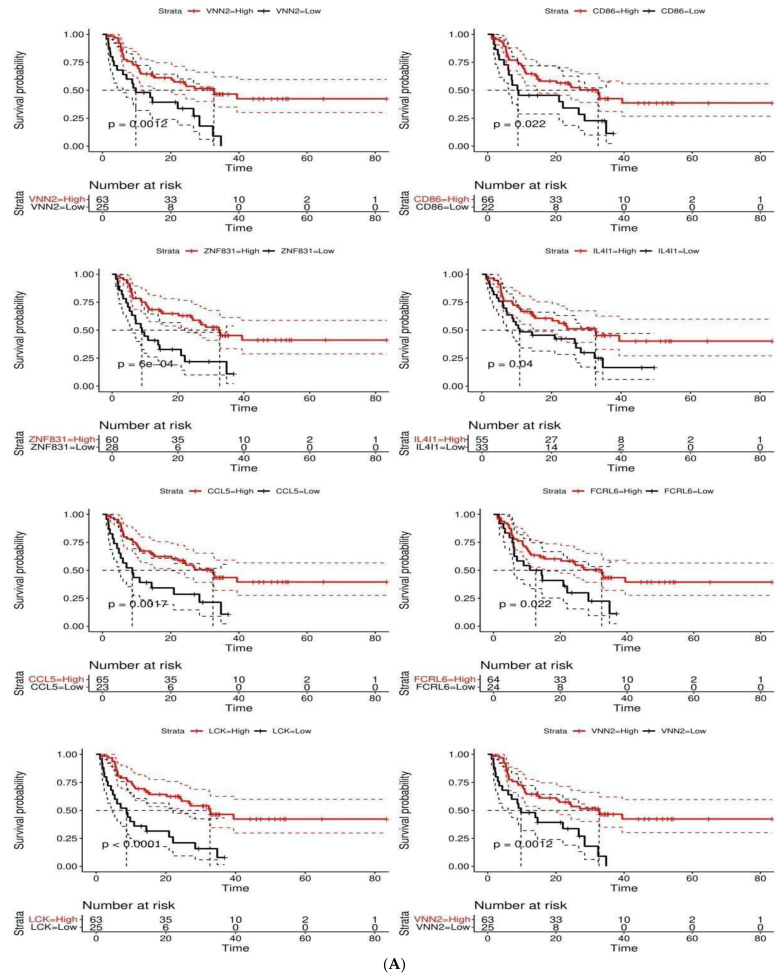
Overall survival value of survival-related genes and RFC16 model: (**A**) Kaplan–Meier plots of overall survival related-genes (CCL5, IL4I1, VNN2, PARP15, LCK, ZNF831, CD86, and FCRRL6). All the genes were significant based on best cutoff value. (**B**) Cox proportional hazard model of the 8 genes. None of the genes showed survival value in the multivariate cox proportional hazard model. (**C**) Feature contribution of RFC16 model.

**Figure 7 cancers-14-05605-f007:**
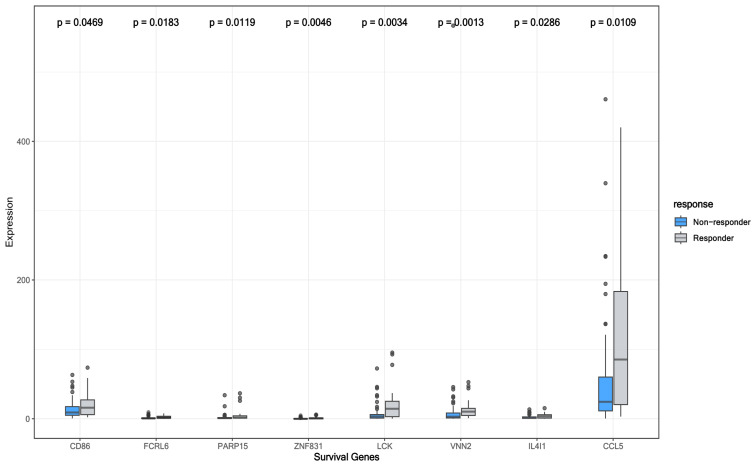
The expression of survival-related genes between responders and non-responders.

**Table 1 cancers-14-05605-t001:** Demographic and clinical characteristics of patients included in the RFC7 model.

Variables	N = 212	Responders (N = 65)	Non-Responders (N = 147)	*p*-Value *
Sex (females), n (%)	68 (32.1)	16 (24.6)	52 (35.4)	0.17
Age (years), median (IQR)	62 (70.3–49)	64 (72–57)	60 (69–47.5)	0.04 *
Histological subtype, n (%)				
Acral Melanoma	5 (2.4)	1 (1.5)	4 (2.7)	1.00
Cutaneous Melanoma	154 (72.6)	37 (56.9)	117 (79.6)	0.001 *
Melanoma of Unknown primary	7 (3.3)	1 (1.5)	6 (4.1)	0.68
Unknown	46 (21.7)	26 (40)	20 (13.6)	0.00
Immunotherapy Type, n (%)				
Anti-CTLA-4	174 (82.1)	44 (67.7)	130 (88.4)	0.001 *
Anti-PD-1	38 (17.9)	21 (32.3)	17 (11.6)	
Overall survival status, n (%)				
Living-1	135 (63.7)	13 (20)	122 (83)	0.00 *
Diseased-0	77 (36.3)	52 (80)	25 (17)	
Log_2_(TMB+1), median (IQR)	3.4 (4.5–2.2)	4.1 (5–3.1)	3.16 (4.3–1.9)	0.00 *
FGA, median (IQR)	0.3 (0.5–0.2)	0.3 (0.5–0.2)	0.4 (0.5–0.2)	0.21
Training set, n (%)	169 (80)	-	-	
Testing set, n (%)	43 (20)	-	-	

*: Statistical significance of *p*-value < 0.05.

**Table 2 cancers-14-05605-t002:** Evaluation metrics scores for each response model.

Model	Mean Bootstrap Estimate	95% CI	10-Fold CV	Precision	Recall	F1-Score	AUC
**RFC7**	0.73	0.60–0.86	0.67	NR: 0.81 R: 0.57	NR: 0.91 R: 0.36	NR: 0.85 R: 0.44	0.71
**TMB alone**	0.67	0.50–0.84	0.61	NR: 0.76 R: 0.43	NR: 0.73 R: 0.46	NR: 0.75 R: 0.44	0.68
**RFC-Seq**	0.75	0.54–0.93	0.67	NR: 0.79 R: 1.0	NR: 1.0 R: 0.38	NR: 0.88 R: 0.55	0.87
**GSTA3 alone**	0.83	0.62–1.00	0.65	NR: 0.75 R: 0.43	NR: 0.79 R: 0.38	NR: 0.77 R: 0.40	0.80
**RFC-Surv**	0.76	0.54–0.91	0.68	NR: 0.78 R: 0.75	NR: 0.95 R: 0.38	NR: 0.86 R: 0.50	0.76
**VNN2 alone**	0.84	0.63–1.00	0.70	NR: 0.76 R: 0.76	NR: 0.89 R: 0.44	NR: 0.82 R: 0.53	0.82
**RFC16**	0.77	0.58–0.92	0.67	NR: 0.77 R: 0.60	NR: 0.89 R: 0.38	NR: 0.83 R: 0.46	0.88

**Table 3 cancers-14-05605-t003:** Univariate and multivariate cox proportional hazard model for OS.

Variable	Category	N (%)	Univariate		Multivariate	
			HR (95% CI)	*p*-Value *	HR (95% CI)	*p*-Value
Age	(>66)	49 (55.7)	-	-	-	-
	(<66)	39 (44.3)	1.09 (0.63–1.88)	0.76	-	-
Sex	Female	34 (38.6)	-	-	-	-
	Male	54 (61.4)	0.61 (0.35–1.06)	0.08	-	-
CD86	High	67 (75.3)	-	-	-	-
	Low	22 (24.7)	1.96 (1.09–3.51)	0.02 *	0.83 (0.36–1.92)	0.67
FCRL6	High	64 (71.9)	-	-	-	-
	Low	25 (28.1)	1.95 (1.09–3.49)	0.03 *	0.61 (0.24–1.56)	0.30
PARP15	High	67 (76.1)	-	-	-	-
	Low	21 (23.6)	2.19 (1.17–4.09)	0.01 *	1.40 (0.60–3.26)	0.43
ZNF831	High	60 (68.2)	-	-	-	-
	Low	28 (31.8)	2.62 (1.48–4.64)	<0.01 *	0.95 (0.29–3.11)	0.94
LCK	High	63 (71.6)	-	-	-	-
	Low	25 (28.4)	3.03 (1.72–5.33)	<0.001 *	4.33 (0.91–20.52)	0.07
VNN2	High	63 (71.6)	-	-	-	-
	Low	25 (28.4)	2.49 (1.41–4.42)	<0.01 *	1.69 (0.76–3.77)	0.20
IL4I1	High	55 (62.5)	-	-	-	-
	Low	33 (37.5)	1.76 (1.02–3.05)	0.04 *	1.03 (0.50–2.21)	0.94
CCL5	High	65 (73.9)	-	-	-	-
	Low	23 (26.1)	2.46 (1.38–4.38)	<0.01 *	0.78 (0.23–2.62)	0.69

*: statistical significane of *p*-value < 0.05.

## Data Availability

Data used in this study are publicly available from the gene expression omnibus (GEO) and cbioportal databases can be accessed through GSE91061 and GSE78220. The study workflow is shown in the Figure 1.

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
