# Peer review of "Genomic and Transcriptomic Predictors of Response to Immune Checkpoint Inhibitors in Melanoma Patients: A Machine Learning Approach"

_cancers, 2022, doi:10.3390/cancers14225605_

Round 1
Reviewer 1 Report
The article utilized multiple melanoma datasets with immune checkpoint inhibitor (ICI) treatment, integrated multi-omics, clinicodemographic factors to build machine learning models to identify the features for ICI response predictors. These ICI-associated biomarkers are also correlated with overall survival and patients’ prognosis. A total of 212 ICI-treated melanoma patients were used for four supervised random forest classification models, all models achieved over 0.7 of AUC and bootstrap estimate. Authors found tumor mutation burden, GSTA3 and VNN2 were the highest contributing features. The levels of gene expression show correlations with immune cells in tumor microenvironment. Overall, this is a retrospective analytical study, the data-driven, hypothesis-generation, and exploratory findings should be experimentally tested or validated using large independent cohort.
There are several defects in this manuscript, which raise the concerns of the results:
Authors defined the responders and non-responders with two different criteria: Durable clinical benefit, and tumor objective response effects (CR, PR, SD and PD). The durable clinical benefit usually is defined as time of 6 months or more to progression from the treatment. Authors should explain how to reconcile the two criteria in this study.
Authors explain why to use |log2FC|>0.5 cutoff for differential expression analysis, while use |log2FC|>2 for correlation with immune cell and stromal cells (line 90 – 107).
Authors used TCGA-SKCM survival-related genes overlapped with SVM-RFE top 100 DEGs ICI-predictors. Most TCGA-SKCM samples are relative early stage with unknown treatment information, while 100 DEGs ICI-predictors are biologically very different, therefore, it is not an ideal experimental design.
Authors should make it clear how to manipulate the confounder factors from different data sets during their analyses. For example, patients from different cohorts (MSKCC (2014) cohort: CTLA4 blockade; DFCI (2015) data cohort: CTLA4 blockade (ipilimumab); UCLA (2016) cohort: PD-1 blockade; GSE91061 (PD1 + CTLA4) blockade; GSE78220 PD-1 blockade) received different ICI treatment, the sequence variations, batch effects et al.
Authors used log2(TMB+1), FGA, mutation count, sex, age at diagnosis, ICI agent, and histological subtype for their RFC7 models, most of them are known to contribute to ICI response, there is now new findings. I expect to see other genomic alterations such as specific gene mutations or copy number changes could predict ICI response.
Authors also make efforts to compare the predictors with other predictors identified in literature.
Other minor comments:
The number of figures cited in the manuscript is disordered.
Fig5B, please add the name of the last column. I assume it is p value.
Author Response
- There are several defects in this manuscript, which raise the concerns of the results:
"Please see the attachment."
- Authors defined the responders and non-responders with two different criteria: Durable clinical benefit, and tumor objective response effects (CR, PR, SD and PD). The durable clinical benefit usually is defined as time of 6 months or more to progression from the treatment. Authors should explain how to reconcile the two criteria in this study.
Response: Thank you for illustrating this point. Revising the included datasets all of them evaluated ICI response using the tumor objective response effects as (CR, PR, SD, and PD) according to the RECIST criteria, thus the durable clinical benefit was not the response measure of effect. In addition to this comment, the methods section page 2 (line 80-81) has been edited as Durable clinical benefit → Tumor objective response.
- Authors explain why to use |log2FC|>0.5 cutoff for differential expression analysis, while use |log2FC|>2 for correlation with immune cell and stromal cells (line 90 – 107).
Response: We thank the reviewer for highlighting this point. Indeed, a higher cut-off was used for immune cell infiltration correlation. This was to take into account the limitation of using an in-silico based approach using bulk RNAseq data to quantify immune cells rather than a more accurate approach using single cell RNAseq or specific protein-markers analysis for tumor samples. Therefore, to limit resulting false positive immune components, we used a different more rigorous cut-off point for |log2FC|.
- Authors used TCGA-SKCM survival-related genes overlapped with SVM-RFE top 100 DEGs ICI-predictors. Most TCGA-SKCM samples are relative early stage with unknown treatment information, while 100 DEGs ICI-predictors are biologically very different, therefore, it is not an ideal experimental design.
Response: We agree with the reviewer that the patients’ characteristics differ between TCGA-SKCM dataset patients and the GEO datasets. However, since the survival rate of early-stage melanoma is above 90%, most survival-related events would be more related to advanced disease. In addition, TCGA-SKCM survival-related genes are prognostic regardless of the treatment. Our aim from overlapping the TCGA-SKCM survival-related genes and SVM-RFE top 100 DEGs ICI-predictors is to select genes with both prognostic and predictive capabilities. We added the following sentence “To select genes with both prognostic and predictive values” to the methods section (lines 121-122) to further clarify this.
- Authors should make it clear how to manipulate the confounder factors from different data sets during their analyses. For example, patients from different cohorts (MSKCC (2014) cohort: CTLA4 blockade; DFCI (2015) data cohort: CTLA4 blockade (ipilimumab); UCLA (2016) cohort: PD-1 blockade; GSE91061 (PD1 + CTLA4) blockade; GSE78220 PD-1 blockade) received different ICI treatment, the sequence variations, batch effects et al.
Response: The type of ICI treatment was considered in the RFC7 model. Our model showed low feature contribution of the type of ICI used. None of the datasets were combined, limiting the presence of batch effect and the need for correction.
- Authors used log2(TMB+1), FGA, mutation count, sex, age at diagnosis, ICI agent, and histological subtype for their RFC7 models, most of them are known to contribute to ICI response, there is now new findings. I expect to see other genomic alterations such as specific gene mutations or copy number changes could predict ICI response.
Response: Thank you for illustrating this point. As you mentioned most of the features included in the clinical RFC7 model are well-validated and proven as ICI biomarkers. However, we also constructed other transcriptomic model with novel genes contributing to ICI response prediction and melanoma prognosis, and the clinical model with the well-evaluated features were constructed to compare the difference in models’ accuracies in ICI response prediction and which of them provides a better predictor, in our case the transcriptomic models outperformed the clinical RFC7 model, which can be translated in a more precise prediction for ICI response, as there are several factors that effect the clinico-genomic features (such as TMB) which includes smoking status and UV exposure in lung cancer and melanoma respectively. As for the copy number alteration, we included the fraction of genome altered (FGA), which represents the fraction of the gained or lost copy number variations over the genome size corrected for tumor purity, ploidy and heterogeneity.1
References:
- Shen R, Seshan VE. FACETS: allele-specific copy number and clonal heterogeneity analysis tool for high-throughput DNA sequencing. Nucleic Acids Res. 2016;44(16):e131. doi:10.1093/nar/gkw520
- Authors also make efforts to compare the predictors with other predictors identified in literature.
Response: We have elaborated on this point in the discussion on page 17 and 18, lines 405 – 421: "The RFC TMB alone model did not perform better in comparison to the RFC7 model which is in agreement with a previous machine learning report in solid cancers where a multifactorial model including both genomic and clinical features outperformed the TMB-based model.19 In their study, Chowell and colleagues developed a machine learning model using 1479 patients across 16 cancer types (37% had non-small cell lung cancer and 13% had melanoma).19 Their 16-variable model which incorporated clinical (age, sex, ICI agent, cancer stage, BMI, neutrophil-to-lymphocyte ratio, cancer type, chemotherapy status, , blood parameters),molecular and genomic (TMB, fraction of copy number alterations, HLA-1 evolutionary divergence, loss of heterozygosity in HLA-1, MSI) has outperformed the established TMB based model (AUC: 0.85 vs 0.62). Other established ICI response biomarkers include CD8+ T cells infiltration levels and PD-L1 expression levels both on tumor and immune cells.20,21 One study further highlighted the prognostic value of the presence of CD137 on circulating CD8+ T cells in resected stage 3 melanoma patients after adjuvant ipilimumab + nivolumab combination therapy.22 Our results should be confirmed in clinical trials settings to assess their performance in comparison to other reported predictors.".
Other minor comments:
The number of figures cited in the manuscript is disordered.
- All required edits have been made.
Fig5B, please add the name of the last column. I assume it is p value.
- The label for the p-value column has been added.
Reviewer 2 Report
Dear Editor and Author, In my opinion, whole paper does not raise any substantive doubts.It is written in clear and transparent scientific language.
All abbreviations are clearly explained.
The methodology, results and conclusions are detailed. As a clinician and researcher, I enjoy studies like that for its possible use in future diagnostics or targeted therapy. I fully recommend accepting the paper for publishing in Cancers. Regards
Author Response
Thank you for your points and illustration on the value of the paper.
Round 2
Reviewer 1 Report
Authors have addressed main issues as a result of significant improvement. It is recommended to submit the revised manuscript for publication.